# Effects of Immunonutrition on Comprehensive Complication Index in Patients Undergoing Pancreatoduodenectomy

**DOI:** 10.3390/medicina56020052

**Published:** 2020-01-24

**Authors:** Jaroslav Tumas, Eugenijus Jasiūnas, Kęstutis Strupas, Audrius Šileikis

**Affiliations:** 1Clinic of Gastroenterology, Nephrourology and Surgery, Institute of Clinical Medicine, Faculty of Medicine, Vilnius University, 03101 Vilnius, Lithuania; kestutis.strupas@santa.lt (K.S.); audrius.sileikis@santa.lt (A.Š.); 2Centre of Informatics and Development, Vilnius University Hospital Santaros Klinikos, 08406 Vilnius, Lithuania; eugenijus.jasiunas@santa.lt

**Keywords:** pancreatic ductal adenocarcinoma, pancreatoduodenal resection, outcomes, pancreatic tumour, nutritional impairments

## Abstract

*Background and objectives:* Immunonutrition is recommended by enhanced recovery after surgery in patients undergoing pancreatoduodenectomy for 5–7 days perioperatively as it may reduce the rate of infectious complications. However, data on effect of immunonutrition on the overall complication rate are contradictory and it is not clear, which groups of patients benefit most. The aims of this study are to evaluate the effects of immunonutrition on the overall complication rate and the rate of severe and/or multiple complications in patients with pancreatic tumours stratified according to final histological diagnosis—patients with pancreatic ductal adenocarcinoma (PDAC) vs. other tumours—and nutritional state, using more sensitive Comprehensive Complication Index. *Materials and Methods:* Seventy consecutive patients scheduled for pancreatoduodenectomy because of pancreatic tumours were randomised into immunonutrition vs. control groups and stratified according to final histological diagnosis and nutritional status. Surgical outcomes were assessed postoperatively using Clavien—Dindo classification (CDC) and Comprehensive Complication Index (CCI). *Results:* No significant differences in the overall complication rates in immunonutrition vs. control, patients with malnutrition vs. no malnutrition, PDAC vs. other pancreatic tumours groups were detected. However, significant differences in the rates of severe and/or multiple complications in immunonutrition vs. control groups and in PDAC patients segregated according to immunonutrition were obtained using CCI. *Conclusions:* Patients with PDAC may experience greater benefits of immunonutrition as compared to patients with benign pancreatic diseases or less aggressive tumours, while nutritional status was not a determining factor for the efficacy of immunonutrition.

## 1. Introduction

Pancreatoduodenectomy represents a technically demanding major abdominal surgery that is associated with significant morbidity [1,2]. Tissue damage results in the release of stress hormones and mediators of systemic inflammation that promote catabolism. Glucose, free fatty acids and amino acids are released from the body’s stores, mainly liver, skeletal muscle and adipose tissue, and are used for acute phase protein synthesis, generation of energy and reconstitution of immune cells, fibroblasts and damaged tissues. Activated immune cells undergo metabolic reprogramming, in many respects resembling the metabolic reprogramming of cancer cells. Glycogen stores are virtually absent in lymphocytes, while metabolic requirements upon activation (both bioenergetic and biosynthetic) increase markedly: cells double in size and are actively proliferating and producing cytokines [3]. Patients with metabolic, nutritional and immune deficiencies (e.g., those with malnutrition and cancer) may be even more vulnerable to major surgical stress and may have limited capacities to recover. Hence, it is very important to assess the risks and take care that the body has sufficient reserves before major surgical stress occurs [4]. In surgery, management of inflammatory responses has a crucial role in improving outcomes [5,6].

Immunonutrition is a method of nutritional management of surgical patients, introduced during the last decade. The main purpose of immunonutrition is to modulate the postoperative inflammatory response with dietary supplements that presumably work on the immune system. Although exact mechanisms of action are still unclear, individual studies have found effects on various immune functions, e.g., immune cell activation [7], concentrations of cytokines and lymphocyte counts [8], and T lymphocyte differentiation [9]. The main components of immunonutrition are arginine, glutamine and polyunsaturated ω-3 fatty acids.

According to the recommendations of the enhanced recovery after surgery (ERAS), patients undergoing pancreatoduodenectomy may receive immunonutrition for 5–7 days perioperatively (evidence level: moderate, recommendation grade: weak) [10]. However, although immunomodulatory effects of immunonutrition have been the subject of a series of clinical research studies [7,8,11,12,13,14,15,16] many unanswered questions remained. The key finding is that immunonutrition may reduce the incidence of infectious complications and the length of hospitalisation. Studies of effects on other parameters (e.g., overall complication rates, postoperative immune functions) produced conflicting results. Complication assessment tools may have played a role in research findings. In the majority of studies to date, the main tool to evaluate complication rates was Clavien–Dindo classification (CDC), however, this classification assesses only the most severe of all complications and may lack sensitivity. The comprehensive complication index (CCI), described in 2013, incorporates all complications and their severity as recorded by the CDC [17].

In addition, it is currently unclear which patient groups are most likely to benefit from immunonutrition. Studies to date included heterogenous and mostly unstratified study groups, e.g., all patients scheduled for pancreatic duodenal resection, patients with any histological type of pancreatic cancer or those with periampular cancer. In many cases, preoperative nutritional state was not taken into consideration. Meanwhile, underlying disease states may significantly affect metabolic, nutritional, and immune systems and eventually, responses to major surgical stress.

The aims of this study are to evaluate the effects of immunonutrition on the overall complication rate and the rate of severe and/or multiple complications in patients with pancreatic tumours undergoing pancreatoduodenectomy. Patients were stratified according to final histological diagnosis and nutritional state, complication rate and severity were assessed using comprehensive complication index.

## 2. Materials and Methods

### 2.1. Patients and Data

Type of the study—prospective, monocentric, randomised. The study was approved by the relevant institutional review boards (Vilnius Regional Biomedical Research Committee permission 2016-01-12 No. 158200-16-810-341, State Data Protection Inspectorate permission 2016-03-21 No. 2R-1807 (2.6-1)). All consecutive patients scheduled for pancreatoduodenectomy due to suspicion of pancreatic cancer at Clinic of Gastroenterology, Nephrourology and Surgery, Institute of Clinical Medicine, Faculty of Medicine, Vilnius University between February 2016 and November 2018 were recruited to the study after giving informed consent (Figure 1). Patients were excluded from the study after enrolment when (1) other pancreatic surgeries were performed instead of pancreatoduodenectomy or (2) patient failed to follow study protocol. All decisions to schedule patients for pancreatoduodenectomy were taken at multidisciplinary team meetings. All surgeries were performed by an experienced pancreatic surgery team, 64 out of 70 surgeries were performed by experienced surgeons (experience of >60 PDR surgeries). All surgeries included standardised pylorus-preserving pancreatoduodenectomy with D1 lymphadenectomy.

Study subjects were randomised into two groups:(1)The first (immunonutrition) group received 5 days of preoperative immunonutrition (L-arginine 6.04 g/day and polyunsaturated fat 4 g/day) in addition to the usual preoperative nutritional management;(2)The second (control) group received a routine preoperative nutritional management only.

Routine perioperative nutritional management included preoperative nutritional screening (NRS-2002) and supplementation with standard normocaloric formula for those at high nutritional risk (NRS ≥ 3) up to 5 days. All patients received infusions of glucose solution in the morning of surgery (200 mL, 5%). At POD1–3, patients got normocaloric enteral formula that was provided at an increasing rate and gradually replaced by oral nutrition at POD4–5 according to the state of a patient.

For each patient, clinical and laboratory testing information was obtained: demographics, medical history, clinical and nutritional evaluation, results of laboratory testing and histological investigation of specimens removed during surgery. Nutritional status was evaluated in every patient one day before the start of immunonutrition in the immunonutrition group and one day before surgery in the control group. Nutritional evaluation included anthropometric measurements, bioelectrical impedance analysis and lumbar skeletal mass index (LSMI) measurements on CT. Bioelectrical impedance analysis (BIA) was performed using InBody S10 according to manufacturer’s recommendations and European Society of Clinical Nutrition and Metabolism (ESPEN) guidelines [18]. High resolution CT images were performed routinely as part of diagnostic investigations in every patient scheduled for pancreatic surgery. Cross-sectional area of muscle was analysed in contrast-enhanced CT scans at the level of the third lumbar vertebra (L3) as described in Baracos et al. 2013 [19]. The following criteria were used to define malnutrition [20]: (1) BMI < 18.5 kg/m^2^, (2) Weight loss (unplanned) >10% at any time, or >5% in the last 3 months together with BMI < 20 kg/m^2^ for age <70 years or <22 kg/m^2^ for age ≥70 years, or FFMI < 15 kg/m^2^ for women or <17 kg/m^2^ for men.

Indicators of systemic inflammation (serum interleukin-6 and C-reactive protein) were evaluated according to a standard procedure one day before the start of immunonutrition in the immunonutrition group, and for all patients prior to the surgery and at the 1st, 3rd and 5th postoperative days (POD).

Surgical outcomes were assessed postoperatively within 30 days after discharge. Clavien–Dindo classification (CDC) was used to evaluate the most severe complication, while comprehensive complication index (CCI) was applied for the longitudinal estimation of all complications. The calculator used was www.assessurgery.com [21]. Severe complications were defined as CDC ≥ 3 and/or CCI > 20.9.

For further analyses, deidentified data were used. The MIDAS archive was used for data capture and storage. The system automatically generated backups and data protection systems.

For comparisons, patients were stratified into the following groups: (1) immunonutrition vs. control group; (2) PDAC vs. other pancreatic tumours group; (3) patients with malnutrition vs. patients without malnutrition.

### 2.2. Statistical Analysis

Sample size calculator RAOSOFT (Raosoft, Inc. Seattle, WA, USA) was used to calculate sample size. Thirty to forty pancreatoduodenectomies are performed at Clinic of Gastroenterology, Nephrourology and Surgery, Institute of Clinical Medicine, Faculty of Medicine, Vilnius University annually. The criterion (test) significance level (test precision) α of 5.0% (α = 0.05) was selected. For selected α, the maximum criterion/test power is 1 − β. In this case, it was equal to 80% (1 − β = 0.80). The minimum sample size was calculated with a significance level of 5% and a power of 80% for 30 subjects in each group (60 subjects in total); the expected sample size was 60–90 subjects. The required sample size of 70 patients was achieved.

Statistical analysis was performed using software: R statistical software package v.3.6.0 (© The R Foundation for Statistical Computing), Rstudio Version 1.2.1335 © 2009–2019 RStudio, Inc., Boston, MA, USA. Interval and ratio variables were described by means and standard deviations (SD), medians and median absolute deviations (MAD), minimum (Min) and maximum (Max) quartiles. Shapiro-Wilk and Kolmogorov-Smirnov (K-S) tests were used to check for normality. Spearman correlation coefficient and Cliff’s Delta effect size were used to assess the strength of the relationships. Statistically significant relationships between two independent groups were tested using the Mann-Whitney U test, statistically significant relationships between three and more independent groups were tested using the Kruskal-Wallis H test. When the obtained data were plotted on a four-field (2 × 2) frequency table and when at least one expected number of observations was less than five, Fisher’s exact criterion and Chi-square (χ^2^) were additionally calculated. The effect size for nominal, bivariate variables was calculated using the Cramer Phi method. Relationships between the groups were rated as statistically significant, when the *p*-value was <0.05 and power of statistical tests was 1 − β = 0.80. Boxplots were used for the graphical comparison of the data. In addition, the means in groups and overall (longer dash) mean are shown on the right side of the figures.

## 3. Results

### 3.1. Characteristics of Patients

Ninety-two patients were included into the study. The patients were randomised to two groups: immunonutrition group, *n* = 40, and the control group, *n* = 52. Four patients were excluded prior to surgery, 18 patients were excluded during the surgery because of the change of scope or type of the procedure (these patients underwent surgeries other than pancreatoduodenectomy). The final composition of study groups: immunonutrition group, *n* = 30, control group, *n* = 40 (Table 1).

More than 30 clinical, nutritional and systemic inflammation indicators were collected for each patient prior to surgery. Immunonutrition group included more subjects with malnutrition, but the difference was not statistically significant (Table 1). Malnutrition was also more frequently identified in males, while differences of phase angle measured by BIA were statistically significant (20.7% of females and 56.7% of males had decreased phase angle, *p* = 0.044). No differences in demographic and presurgical nutritional parameters were observed in groups of PDAC vs. other pancreatic tumours.

The final diagnosis of pancreatic ductal adenocarcinoma (PDAC; *n* = 39) or other pancreatic tumours (*n* = 31) was obtained after histological investigation of surgical tissues. Other pancreatic tumours included: periampullary carcinoma (*n* = 11), chronic pancreatitis (*n* = 8), neuroendocrine pancreatic cancer (*n* = 5), intraductal papillary mucinous neoplasm (n = 2), pseudopapillary solid tumour (*n* = 1), mucinous pancreatic adenocarcinoma (*n* = 1), acinar cell pancreatic cancer (*n* = 1), metastatic cancer (renal cellular carcinoma) (*n* = 1), and pancreatic microcystic adenoma (*n* = 1). For further analyses, the patients were stratified into groups of PDAC vs. other pancreatic tumours.

Important and statistically significant differences of cytokine concentrations were observed in groups of PDAC vs. other pancreatic tumours: patients with PDAC had higher indicators of systemic inflammation prior to surgery (mean plasma IL-6 concentration was 5.33 ng/L in the PDAC group and 3.49 ng/L in other pancreatic tumours group, *p* = 0.02, Kruskal-Wallis H test). Postoperatively, PDAC patients had statistically significantly lower indicators of systemic inflammation: mean plasma IL-6 concentration was 117.7 ng/L in the PDAC group and 177.33 ng/L in other pancreatic tumours group, *p* = 0.026, Kruskal-Wallis H test; mean plasma CRP concentration was 147.51 mg/L in the PDAC group and 180.89 mg/L in other pancreatic tumours group, *p* = 0.025, Kruskal-Wallis H test. No statistically significant differences of systemic inflammation were observed in the immunonutrition vs. control groups and in patients with or without malnutrition.

### 3.2. Surgical Outcomes

Overall, 81.4% of patients suffered postoperative complications. Of these, 44.29% (*n* = 24) patients over the period of 30 postoperative days experienced severe and/or multiple complications (CCI > 20.9). 47.1% (*n* = 33) patients had mild complications (CDC1–2; CCI < 20.9).

There were no statistically significant differences in the overall complication rates in immunonutrition vs. control groups (median CCI = 20.9 in both groups; Figure 2).

In PDAC patients segregated according to immunonutrition, median CCI was 8.7 in the immunonutrition group (*n* = 17), while median CCI was 20.9 in the control group (*n* = 22); however, the difference was not statistically significant (*p* = 0.2) (Figure 3).

Evaluation of severe and/or multiple complications presented diverging results. Overall, 18.6% (*n* = 13) had CDC ≥ 3:13.3% (*n* = 4) of patients in the immunonutrition group and 22.5% (*n* = 9) in the control group (Table 2); the difference between the groups was not statistically significant (*p* = 0.333).

Overall, 34.3% (*n* = 24) of patients had CCI > 20.9:23.3% (*n* = 7) of patients in the immunonutrition group and 42.5% (*n* = 17) in the control group; the difference between the groups was statistically significant (McNemar’s chi-squared = 7.5, *p* = 0.006) (Figure 4). Even more significant differences of severe and/or multiple complication rates were obtained when PDAC patients were segregated into immunonutrition (*n* = 17) vs. control (*n* = 22) groups: CCI > 20.9 was found in 17.65% of patients in the immunonutrition group and in 45.45% of patients in the control group (McNemar’s chi-squared = 4.3, *p* = 0.04) (Figure 5).

There were no statistically significant differences of complication rates in patients with or without malnutrition (median CCI = 20.9 in both groups) and in patients at an increased nutritional risk (NRS ≥ 3). Moreover, no statistically significant differences were obtained after segregation of these patients into immunonutrition vs. control groups. Interestingly, statistically significantly higher rates of severe and/or multiple complications were observed in male patients (median CCI = 8.7 in females vs. median CCI = 20.9 in males, *p* = 0.009). Although there were no gender differences in postoperative systemic inflammation indicators, malnutrition was more frequent in male patients.

## 4. Discussion

Enhanced recovery after surgery (ERAS) programmes are multimodal strategies that aim to attenuate the loss of, and improve the restoration of, functional capacity after surgery. According to ERAS recommendations, patients undergoing pancreatoduodenectomy may receive immunonutrition for 5–7 days perioperatively (evidence level: moderate, recommendation grade: weak) [10]. However, it is currently unclear which patient groups are most likely to benefit from immunonutrition (e.g., those with PDAC vs. less aggressive pancreatic diseases or those with nutritional impairments vs. nutritionally normal). The only meta-analysis investigating the effects of immunonutrition on patients undergoing pancreatoduodenectomy was recently published. Immunonutrition was found to reduce the rate of infectious complication and length of hospitalisation, but had no effect on the overall complication rates, non-infectious complication rates, and postoperative mortality. Patients were not stratified according to diagnosis or nutritional state [11]. Several randomised clinical studies included patients undergoing pancreatoduodenectomy and those with pancreatic or periampular cancers. These patient groups were also included into more general studies investigating patients undergoing gastrointestinal surgeries or those with gastrointestinal cancers (Table 3).

In meta-analyses and systematic reviews evaluating the impact of immunonutrition on patients undergoing gastrointestinal surgery, two studies found an effect on rate of infectious complications and length of hospitalisation [26,27], while in one meta-analysis with the largest number of subjects effect on the overall complication rates was also identified [28]. Three meta-analyses and systematic reviews investigated the effects of immunonutrition in patients with various gastrointestinal cancers; in two of them, immunonutrition had effects not only on the rates of infectious complications and length of hospitalisation, but also on the overall complication rates [29,30]. Immunonutrition was found to be cost-effective in surgical patients with gastrointestinal cancers (i.e., this intervention may reduce treatment costs because of the lower complication rates) [24].

Importantly, CDC was used for complication rating in all these studies. The CDC is an excellent and easy to apply system for grouping and grading complications, that is validated and used worldwide. However, while CDC assesses only one of the most severe complications, CCI enables longitudinal estimation of all postoperative complications over a certain period of time [21]. This tool may be especially sensitive in major and complex surgeries that are followed by prolonged recovery periods [32]. In this study, CCI enabled detection of significant effects of immunonutrition on the rate of severe and/or multiple complications in patients with PDAC, undergoing pancreatuduodenal resection, while the overall complication rate measured by either CDC or CCI did not differ between the groups. Although there are no studies of CCI usage in the evaluation of immunonutrition effects on complication rates, CCI demonstrated increased sensitivity and superiority over traditionally reported morbidity endpoints in a recent clinical trial on patients undergoing pancreatoduodenectomy [33]. CCI was also successfully used to evaluate postoperative complication rate in patients undergoing laparoscopic vs. open pancreatoduodenectomy [34]. In several other studies CCI was found to be more strongly correlated with the length of hospitalisation in gastric surgery patients [17] and a valid tool to assess the overall burden of complications in patients undergoing HIPEC treatment [35].

In this study, patients were stratified according to the diagnosis (PDAC vs. other pancreatic tumours) and presence of malnutrition. Interestingly, significant effects of immunonutrition on the rate of severe and/or multiple complications were observed in patients with PDAC, while nutritional status was not a determining factor for the efficacy of immunonutrition. Importantly, PDAC patients displayed significant preoperative and postoperative disturbances of the indicators of systemic inflammation. Metabolic and immune reprogramming are important hallmarks of pancreatic cancer. Cancerous tissue is characterised by a complex and dynamic secretion of various pro-inflammatory and anti-inflammatory cytokines that co-modulate the microenvironment promoting carcinogenesis and metastasis [36,37]. One of the best-studied cytokines with carcinogenesis-promoting activity is IL-6, whose secretion is mediated, among other factors, by activation of the Kras signalling pathway and hypoxic microenvironment [38]. According to the results of this and other studies (Table 3), patients with PDAC may experience greater benefits of immunonutrition as compared to patients with benign pancreatic diseases or less aggressive tumours.

Effect of immunonutrition on patients with nutritional impairments vs. nutritionally normal is also unclear. In a study by Braga et al. the highest clinical benefits were observed in patients at a high risk or with an established malnutrition [4], several studies by Klek et al. also identified benefits of immunonutrition in patients with malnutrition [25,31]. Martin et al. found that patients receiving preoperative immunonutrition had a lower risk of malnutrition and a lower reduction in serum albumin after surgery [15,20]. However, in a study by Silvestri et al. immunonutrition reduced rate of infectious complications and length of hospitalisation in patients without any nutritional impairments [13], whereas Hübner et al. did not identify any differences in patients with malnutrition vs. those without nutritional impairments [9].

The major limitation of this study is a small sample size, hence, any observations need confirmation through testing of larger patient populations. In this study, unselected group of patients under routine clinical setting was investigated and comprehensive clinical, laboratory and imaging data were collected on each subject.

## 5. Conclusions

Patients with PDAC may experience greater benefits of immunonutrition as compared to patients with benign pancreatic diseases or less aggressive tumours, while nutritional status was not a determining factor for the efficacy of immunonutrition.

## Figures and Tables

**Figure 1 medicina-56-00052-f001:**
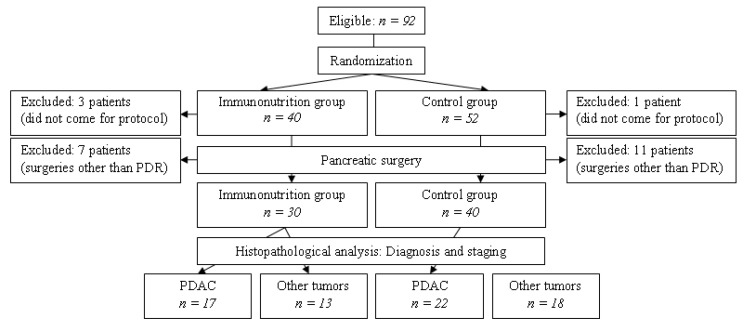
Study flow chart.

**Figure 2 medicina-56-00052-f002:**
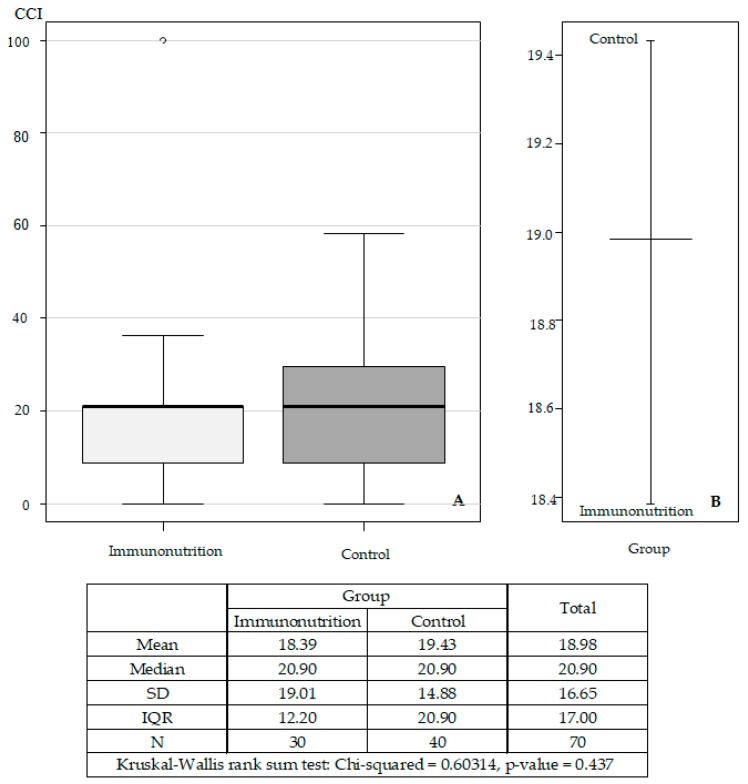
Standard deviations (**A**) and mean (**B**) of the comprehensive complication index (CCI) in the immunonutrition and control groups (*n* = 70).

**Figure 3 medicina-56-00052-f003:**
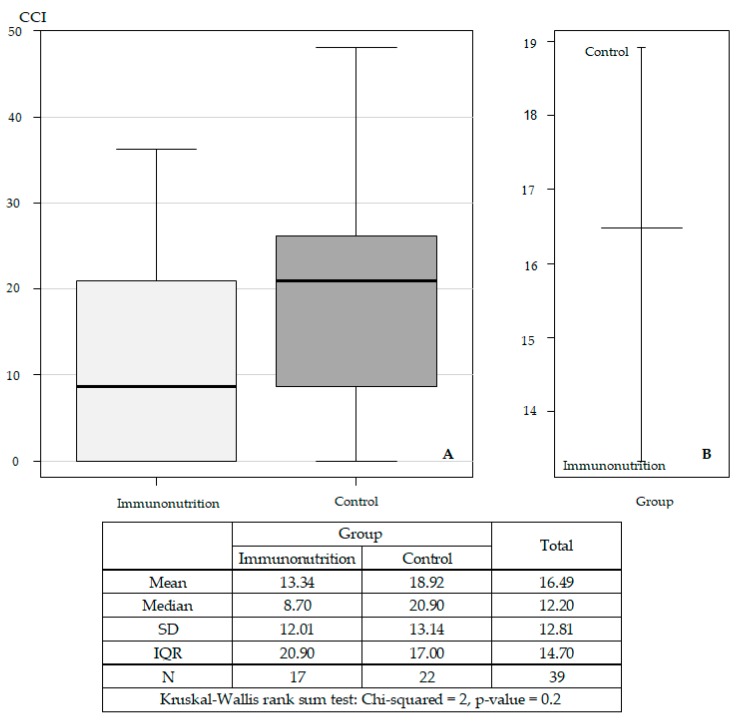
Standard deviations (**A**) and mean (**B**) of the comprehensive complication index (CCI) in PDAC patients: immunonutrition vs. control groups (*n* = 39).

**Figure 4 medicina-56-00052-f004:**
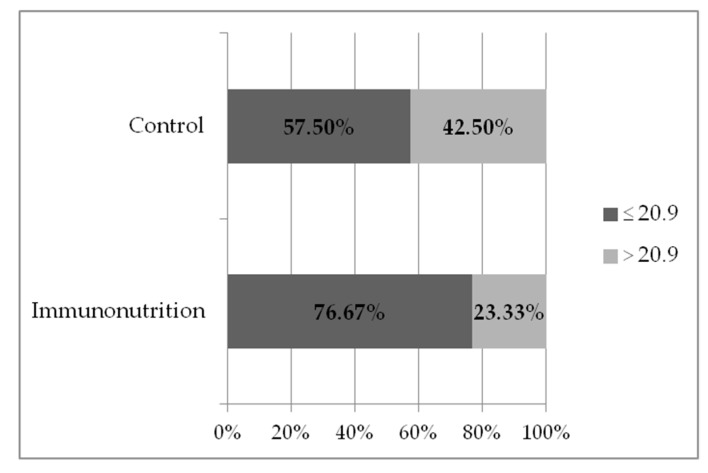
Rate of severe and/ or multiple complications (CCI > 20.9) in immunonutrition vs. control groups (*n* = 70) (*p* = 0.006).

**Figure 5 medicina-56-00052-f005:**
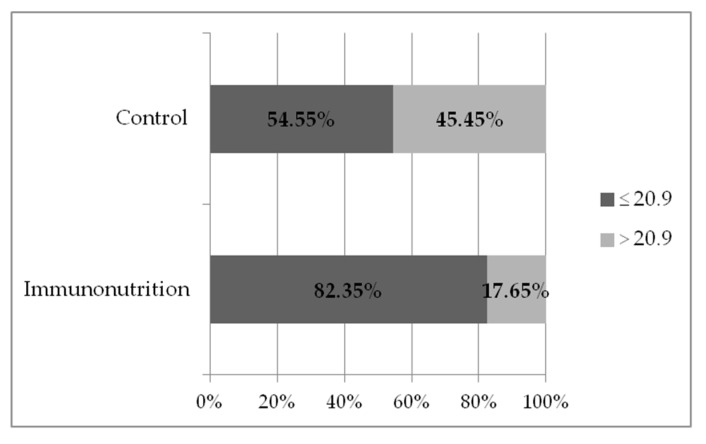
Rate of severe and/or multiple complications (CCI > 20.9) in PDAC patients, stratified to immunonutrition vs. control groups (*n* = 39) (*p* = 0.04).

**Table 1 medicina-56-00052-t001:** Characteristics of patients.

Group	Immunonutrition		Control		Overall		Mann-Whitney
	*n* = 30		*n* = 40		*n* = 70		U test
Statistics	Mean (SD)	Median (MAD)	Mean (SD)	Median (MAD)	Mean (SD)	Median (MAD)	U
							*p*-Value
							Effect Size R
**Age** (years)	62.6 (10.5)	61.5 (88.2)	63.0 (8.7)	61.5 (88.2)	62.8 (9.43)	61.5 (88.2)	U = 596.00
							*p* = 0.962
							*r* = −0.006
**Body mass index** (kg/m²)	26.8 (5.6)	27.2 (5.3)	26.9 (4.2)	26. 5 (3.9)	26.9 (4.79)	26.6 (4.3)	U = 588.50
							*p* = 0.891
							*r* = −0.016
**Weight loss** (kg)	5.3 (6.8)	0.0 (0.0)	5.6 (7.3)	1.6 (2.4)	5.5 (7.04)	0.0 (0.0)	U = 591.50
							*p* = 0.914
							*r* = −0.013
**Interleukin 6** (ng/L)	4.4 (4.2)	2.3 (0.4)	5.6 (6.3)	2.8 (1.2)	5.0 (5.4)	2.6 (0.9)	U = 359.00
							*p* = 0.582
							*r* = −0.073
							Cramer’s φ effect size *p*-value
**Male gender** %	56.7%		50.0%		52.9%		Phi = 0.0661
							*p* = 0.7558
**Malnutrition** %	33.3%		22.5%		27.1%		Phi = 0.1206
							*p* = 0.4167
**Diagnosis PDAC** %	56.7%		55.0%		55.7%		Phi = 0.0166
							*p* = 1.000

**Table 2 medicina-56-00052-t002:** Overall complication rate (Clavien–Dindo classification) in immunonutrition vs. control groups.

Clavien–Dindo Classification	Group				Total	
	Immunonutrition		Control			
Grade	Count	%	Count	%	Count	% of Total
0	6	20.0%	7	17.5%	13	18.6%
1	10	33.3%	15	37.5%	25	35.7%
2	10	33.3%	9	22.5%	19	27.1%
3a	2	6.7%	1	2.5%	3	4.3%
3b	1	3.3%	7	17.5%	8	11.4%
4a	0	0.0%	1	2.5%	1	1.4%
5	1	3.3%	0	0.0%	1	1.4%
Total	30	100.0%	40	100.0%	70	100.0%

**Table 3 medicina-56-00052-t003:** Overview of published data on the use of immunonutrition in various patient groups.

Publication	Number of Patients	Patient Population	Study Design	Outcome Measures	Study Results
Miyauchi Y, 2019 [12]	60	Pancreato-duoden-ectomy	Prospective, randomised. Perioperative or preoperative immunonutrition.	Immune functions, rate of postoperative complications.	No significant differences between the groups. RR 0.76 [0.46–1.28]
Silvestri S, 2016 [13]	54	Pancreato-duoden-ectomy; patients without malnutrition	Case-control. Immunonutrition preoperatively.	Mortality, overall complication rate, rates of individual complications, length of hospitalisation.	Lower rate of infectious complications and shorter duration of hospitalisation in the immunonutrition group. RR 0.87 [0.56–1.36]
Suzuki D, 2010 [7]	30	Pancreato-duoden-ectomy	Randomised, three branches: perioperative immunonutrition, postoperative immunonutrition, control.	Immune functions; rate of infectious complications.	Statistically significant differences of immune functions and rates of infectious complications, RR 0.29 [0.08–1.05] in comparisons of perioperative immunonutrition vs. other groups.
Gade J, 2016 [14]	35	Pancreatic cancer	Randomised case-control.	Rate of postoperative complications, length of hospitalisation, changes of body weight and general clinical status.	No significant differences between the groups. RR 0.70 [0.51–0.95]
Martin RC, 2017 [15]	71	Pancreatic cancer	Randomised case-control. Preoperative immunonutrition.	Overall complication rate and rate of infectious complications, length of hospitalisation, risk of malnutrition postoperatively, serum albumin.	Lower rate of postoperative complications, RR 0.50 [0.24–1.05], shorter duration of hospitalisation, lower risk of malnutrition and less of a decrease of serum albumin in the immunonutrition group.
Hamza N, 2015 [8]	37	Periampular tumours	Randomised case-control. Perioperative immunonutrition.	Immune functions.	Statistically significant differences of immune functions in the immunonutrition group. RR 0.83 [0.32–2.15]
Guan H *, 2019 [11]	299	Pancreato-duoden-ectomy	Meta-analysis; four randomised clinical trials included.		Immunonutrition decreases rate of infectious complications, RR 0.58 [0.37–0.92] and length of hospitalisation; no effect on the overall complication rate, RR 0.81 [0.62–1.05], rate of non-infectious complications, RR 0.94 [0.69, 1.28] and postoperative mortality.
Hübner M, 2012 [9]	152	Gastro-intestinal surgery	Randomised case-control, preoperative immunonutrition, patients with malnutrition.	Rate of postoperative complications, infectious complications, length of hospitalisation.	No significant differences between the groups. RR 0.95 [0.76–1.19]
Burden S *, 2012 [22]	1585	Gastro-intestinal surgery	Meta-analysis; thirteen clinical trials included.		Immunonutrition decreases the overall complication rate, RR 0.67 [0.53–0.84], and rate of infectious complications.
Hegazi RA *, 2014 [23]	1456	Gastro-intestinal surgery	Meta-analysis and systematic review; immunonutrition vs. standard nutritional management and immunonutrition vs. control (no nutritional management). 17 clinical trials included.		Immunonutrition and standard nutritional management decreases rate of infectious complications, OR 0.49 [0.29–0.83] and length of hospitalisation. No significant differences between immunonutrition and standard nutritional management.
Reis AM *, 2016 [24]		Gastro-intestinal surgery	Systematic review; cost-effectiveness of immunonutrition. Six randomised clinical trials included.		Immunonutrition may reduce costs of treatment due to decreased rate of complications.
Klek S (a), 2014 [25]	776	Gastro-intestinal surgery	Randomised clinical trial; enteral and parenteral immunonutrition. Patients with or without malnutrition.	Rate of postoperative complications, length of hospitalisation.	No significant differences in patients without malnutrition. Statistically significant differences in patients with malnutrition when enteral immunonutrition is given, but no differences with parenteral immunonutrition.
Wong CS *, 2016 [26]	2016	Gastro-intestinal surgery	Systematic review; 19 randomised clinical trials included.		Immunonutrition decreases rate of infectious complications and length of hospitalisation; no effect on the overall complication rate and postoperative mortality.
Marimuthu K *, 2012 [27]	2496	Gastro-intestinal surgery	Meta-analysis; 26 randomised clinical trials included.		Immunonutrition decreases rate of infectious complications, RR 0.64 [0.55–0.74] and length of hospitalisation; no effect on the overall non-infectious complication rate, RR 0.82 [0.71–0.95] and postoperative mortality.
Mazaki T *, 2015 [28]	7572	Gastro-intestinal surgery	Meta-analysis. Comparison of enteral and parenteral immunonutrition, enteral and parenteral standard nutritional management. 74 clinical trials included.		Enteral immunonutrition is the most effective in decreasing overall complication rate, OR 0.75 [0.58–0.95], postoperative mortality, rates of wound infections, intraabdominal abscess and sepsis. Parenteral immunonutrition is the most effective in decreasing rates of pneumonia and urinary tract infections. The worst outcomes are obtained with standard parenteral nutritional management.
Yan X *, 2016 [29]	3854	Gastro-intestinal cancers	Meta-analysis; 30 randomised clinical trials included.		Enteral immunonutrition decreases rates of infectious, RR 0.69 [0.48–0.98] and non-infectious complications, RR 0.72 [0.61–0.84], length of hospitalisation.
Song GM *, 2015 [30]		Gastro-intestinal cancers	Meta-analysis, systematic review; 27 randomised clinical trials included.		Immunonutrition pre-, peri- or postoperatively decreases rate of infectious complications, RR 0.58 [0.43–0.78]. Besides, perioperative immunonutrition decreases rate of non-infectious complications, perioperative or postoperative immunonutrition decreases length of hospitalisation.
Adiamah A *, 2019 [16]	1387	Gastro-intestinal cancers	Meta-analysis, systematic review; 16 randomised clinical trials included.		Immunonutrition decreases rate of infectious complications, OR 0.52 [0.38, 0.71] and length of hospitalisation, no effect on the rate of non-infectious complications, OR 0.98 [0.73, 1.33] and postoperative mortality.
Klek S (b), 2010 [31]	305	Gastro-intestinal cancers	Randomised clinical trial. Postoperative immunonutrition, patients with malnutrition.	Rate of postoperative complications, length of hospitalisation, postoperative mortality.	Immunonutrition decreases rate of infectious complications, OR 0.84 [0.42–1.69] and overall complication rate, OR 0.67 [0.35–1.27], length of hospitalisation and postoperative mortality.

* Meta-analyses and systematic reviews.

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
