# Peer review of "Effects of Immunonutrition on Comprehensive Complication Index in Patients Undergoing Pancreatoduodenectomy"

_medicina, 2020, doi:10.3390/medicina56020052_

Round 1

Reviewer 1 Report

The manuscript by Tumas et al evaluated the effects of immunonutrition on the overall complication rate and the rate of severe and/or multiple complications in patients receiving pancreatoduodenectomy because of pancreatic tumours. This study is interesting. I have the following minor concerns:

The immunonutrition group received 5 days of preoperative immunonutrition. But the control group received a routine preoperative nutritional management only. Did the control group receive any other nutrition treatment (placebo) other than routine preoperative nutritional management? How many patients were enrolled and how many opted out because of exclusion criteria?Could the authors include a Flowchart to show it? The authors compared the outcomes of those with or without malnutrition. Did they compared the outcomes of those with or without nutrition risk evaluated by NRS2002? Is the pancreatoduodenectomy performed by one surgeon or by different surgeons? If it is conducted by different surgeons, how to balance the bias? Please revise the format of Table 1 to make it professional. In Table 3, the authors showed the overview of published data on the use of immunonutrition in various patient groups. Can they do the subgroup analysis and include the HR values of immunonutrition for short term outcomes of pancreatoduodenectomy? Please include more information about Enhanced Recovery After Surgery in the discussion.

Author Response

The manuscript by Tumas et al evaluated the effects of immunonutrition on the overall complication rate and the rate of severe and/or multiple complications in patients receiving pancreatoduodenectomy because of pancreatic tumours. This study is interesting. I have the following minor concerns:

The immunonutrition group received 5 days of preoperative immunonutrition. But the control group received a routine preoperative nutritional management only. Did the control group receive any other nutrition treatment (placebo) other than routine preoperative nutritional management?

Routine perioperative nutritional management was described in the section 2.1 Patients and data. This study was not placebo controlled.

How many patients were enrolled and how many opted out because of exclusion criteria? Could the authors include a Flowchart to show it?

The study flow chart was included into the manuscript as Figure 1.

The authors compared the outcomes of those with or without malnutrition. Did they compared the outcomes of those with or without nutrition risk evaluated by NRS2002?

Indeed, we compared outcomes in patient groups with increased nutritional risk or nutritional impairments vs. those without and didnot find statistically significant differences. We supplemented the manuscript with this information. Besides, we acknowledge that small sample size is the major limitation of this study that may potentially preclude identification of differences in subgroups.

Is the pancreatoduodenectomy performed by one surgeon or by different surgeons? If it is conducted by different surgeons, how to balance the bias?

As Vilnius University Hospital Santaros Klinikos is one of the two major tertiary level university hospitals in Lithuania, it covers approximately half of the population for pancreatic surgery (population of Lithuania is ~2,9 millions). A dedicated team of surgeons comprise a pancreatic surgery group and participates in multidisciplinary team meetings. All surgeries in this study were performed by this dedicated group and 64 out of 70 surgeries were performed by experienced surgeons (defined as having performed > 60 PDR surgeries). Efforts to standardize procedures and follow quality improvement rules are implemented where possible in our hospital. Hence, we believe that we diminish impact of human factor to the minimal possible.

Please revise the format of Table 1 to make it professional.

The table was revised accordingly.

In Table 3, the authors showed the overview of published data on the use of immunonutrition in various patient groups. Can they do the subgroup analysis and include the HR values of immunonutrition for short term outcomes of pancreatoduodenectomy?

Unfortunately, we could not perform calculations of hazard ratio (HR) in this study as HR has to take into account not only the total number of events, but also of the timing of each event, and this information was not provided in the studies described in Table 3. In order to improve the presentation of results in Table 3, we included risk ratio (RR) and odds ratio (OR) for every study and outcome, where it was available. Hopefully, this has improved the presentation and interpretation of the effect of immunonutrition on various outcomes. Furthermore, we do not claim that this study collected information on all available randomized clinical trials according to the rules applied to meta-analysis (e.g., PRISMA), although we tried to calculate RR and OR for the overall complication rate based on studies in Table 3 to identify trends (some results below). Nevertheless, we believe that we were able to collect all meta-analyses and systematic reviews relevant to this study.

Please include more information about Enhanced Recovery After Surgery in the discussion. 

Thank you for the suggestion, we included information on ERAS into Discussion to give an introduction to the main issues that were investigated and discussed in this manuscript.

Reviewer 2 Report

In this article, the authors compared the effects of immunonutrition in patients undergoing pancreatoduodenectomy.  The paper is well written and thorough. Below are few minor comments - 

Please add references to line 62 - "However, although immunomodulatory effects of ........ series pf clinical research studies (add reference), many unanswered....." Figures 1 and 2 y-axis needs to be corrected. Figures in figure 1 and 2 can be labeled as "A", "B" and explained accordingly in the figure legends and in the statistical analysis section

Author Response

In this article, the authors compared the effects of immunonutrition in patients undergoing pancreatoduodenectomy.  The paper is well written and thorough. Below are few minor comments - 

Please add references to line 62 - "However, although immunomodulatory effects of ........ series pf clinical research studies (add reference), many unanswered....."

The references were added accordingly.

Figures 1 and 2 y-axis needs to be corrected. Figures in figure 1 and 2 can be labeled as "A", "B" and explained accordingly in the figure legends and in the statistical analysis section.

Thank you for the suggestion, figures were corrected accordingly.